# Evidence for the oxidation of Earth's crust from the evolution of manganese minerals

Daniel R. Hummer [1✉], Joshua J. Golden[2], Grethe Hystad[3], Robert T. Downs[2], Ahmed Eleish[4], Chao Liu[5], Jolyon Ralph[6], Shaunna M. Morrison [5], Michael B. Meyer[5] & Robert M. Hazen[5]

Analysis of manganese mineral occurrences and valence states demonstrate oxidation of Earth's crust through time. Changes in crustal redox state are critical to Earth's evolution, but few methods exist for evaluating spatially averaged crustal redox state through time. Manganese (Mn) is a redox-sensitive metal whose variable oxidation states and abundance in crustal minerals make it a useful tracer of crustal oxidation. We find that the average oxidation state of crustal Mn occurrences has risen in the last 1 billion years in response to atmospheric oxygenation following a 66 ± 1 million-year time lag. We interpret this lag as the average time necessary to equilibrate the shallow crust to atmospheric oxygen fugacity. This study employs large mineralogical databases to evaluate geochemical conditions through Earth's history, and we propose that this and other mineral data sets form an important class of proxies that constrain the evolving redox state of various Earth reservoirs.

[1] School of Earth Systems and Sustainability, Southern Illinois University, Carbondale, IL 62901, USA. [2] Department of Geosciences, University of Arizona, Tucson, AZ 85721, USA. [3] Department of Mathematics and Statistics, Purdue University Northwest, Hammond, IN 46323, USA. [4] Tetherless World Constellation, Rensselaer Polytechnic Institute, Troy, NY 12180, USA. [5] Earth and Planets Laboratory, Carnegie Institution for Science, Washington, DC 20015, USA. [6] Mindat.org, 128 Mullards Close, Mitcham, Surrey CR4 4FD, UK. ✉email: daniel.hummer@siu.edu

The oxygenation of Earth's atmosphere by photosynthesizing cyanobacteria is widely regarded as one of the most influential processes in Earth's 4.5-billion-year history. Not only did the accumulation of free $O_2$ directly enable the evolution of modern animals[1], but it also fundamentally changed Earth's near-surface mineralogy, petrology, and geochemical cycles[2–11]. Numerous proxies have been used to help reconstruct past concentrations of atmospheric $O_2$, including carbon[2,3] and sulfur[4–7] isotope compositions, trace element concentrations in oceanic sediments[8–11], and organic biomarkers[12,13]. Using these and other approaches, reconstructions of atmospheric oxygen have been assembled for at least the most recent 2.5 billion years of Earth history[14–17] if not more, and these efforts have recently been summarized in reviews by Lyons et al.[18] and Canfield[19]. However, the complex relationships between oxygenation of the atmosphere and the co-evolution of Earth's lithosphere, hydrosphere, atmosphere, and biosphere are still poorly understood[18]. In this contribution, we report on a revealing proxy for the redox state of Earth's shallow crust using large mineralogical data resources and we assess implications for the co-evolution of Earth's geosphere and biosphere.

The current consensus is that near 2.5 Ga, the production of $O_2$ by photosynthetic cyanobacteria began to outweigh various geochemical sinks of $O_2$, causing molecular oxygen to permanently accumulate in the atmosphere[4]. This unique period in geologic history, now acknowledged as the initial stage of an ongoing process rather than as a single event, is nevertheless referred to as the "Great Oxidation Event" (GOE)[20]. The evolution of oxygenic photosynthesis must have taken place before the GOE, although the question of whether it occurred immediately before[21] versus over the prior one billion years[11] is still debated. Most researchers favor a scenario in which photosystem-II, the dominant mechanism of photosynthesis on modern Earth, evolved significantly before the GOE and in the intervening time caused temporary "whiffs" of oxygen[8,22,23] to appear in the geologic record before $O_2$ concentrations were depleted back to well below $10^{-5}$ of the present atmospheric level (PAL)[14] by various biological and geochemical oxygen sinks[24–26]. After a significant build-up of free $O_2$ in the atmosphere, followed by a possible downturn[18], oxygen levels became mostly static for nearly 2 billion years[27,28]. This prolonged period of stasis was followed by a notable increase near 600 Ma[29], which then fluctuated somewhat before reaching PAL[14–17] (see Fig. 1 of Ref. [18] for a summary of these changes).

The increase in atmospheric and oceanic $O_2$ content following the GOE led to the widespread oxidation of redox-sensitive metals[17,30], with a corresponding increase in Earth's mineral diversity. For example, Hazen et al.[31] estimate that more than half of known mineral species are the result of oxidative weathering of preexisting minerals. It is unlikely that the consequent increase in mineral diversity, as well as the change in redox state of abundant crustal transition metals such as Fe, Mn, and Co caused by the GOE, left no signature in the geologic record. In particular, the occurrence and distribution of Mn minerals through geologic time is a useful proxy for the redox state of Earth's crust (in terms of its average $fO_2$) due to its high crustal abundance (~1000 ppm)[32], lithophilic character, presence in 560 distinct mineral species, and three naturally occurring oxidation states of +2, +3, and +4, all of which occur in mineral species.

Although recent work has shed light on the oxidation of Earth's mantle via the hydrothermal oxidation of oceanic basalt and its subsequent subduction into mantle wedges at plate boundaries[33,34], relatively few data are available on the evolving redox state of Earth's crust. Mineralogical markers, notably their redox-sensitive transition elements, provide one possible solution. For example, Golden et al.[35] tabulated the rhenium content of

422 molybdenite specimens from the past 3 billion years and documented a systematic increase in crustal oxidation state—an increase that lagged behind the rise in atmospheric oxygen by hundreds of millions of years. The results of the molybdenite study demonstrate that the behavior of redox-sensitive metals can be used to trace the redox state of Earth's subsurface reservoirs through geologic time.

Here we employed mindat.org and the Mineral Evolution Database (MED at rruff.info/ima), focusing on Mn mineral occurrences as of 20 November 2015. We recorded 22,064 mineral-locality data pairs (i.e., a single mineral species occurring at a single locality), 2666 of which had associated geologic ages in the MED. In these data sets and the discussion that follows, a "Mn mineral" is considered to be any mineral species approved by the International Mineralogical Association that contains Mn as an essential element in its nominal chemical formula. Although some crustal Mn is contained as a minor constituent in other minerals, this Mn is nearly all in the form of $Mn^{2+}$ (substituting for $Fe^{2+}$)[36] and would therefore not affect our analysis of the emergence of more oxidized Mn mineral species. Likewise, while ocean floor Mn nodules constitute a significant reservoir of present-day Mn and are predominantly $Mn^{3+}$ and $Mn^{4+}$, these are geologically recent and would not alter the temporal trend discussed below.

## Results and discussion

A histogram of dated Mn mineral occurrences using bins of 50 Ma (Fig. 1) shows that maxima in Mn mineralization occur in close proximity to five episodes of supercontinent formation. This trend is well known for detrital zircon[37,38] and is observed in other mineral groups, such as Hg, Cu, Co, and Be minerals[39–41]. The close association between mineralization events and the formation of supercontinents is understood as the result of convergent margins, where continental crust melts and creates new pressure-temperature regimes in which various forms of primary mineralization, such as crystallization in igneous rock or hydrothermal veins, can occur. Patterns of Mn mineralization are somewhat different during the period of Rodinia, which features peaks before and after but very few during the supercontinent's existence. This unusual trend may be due to unique tectonic features of Rodinian assembly, which led to enhanced erosion of orogenic mineral deposits for low field strength elements such as Mn, while enhancing those for high field strength elements such as Y, Nb, and Zr[42].

In addition to the frequency of mineralization events, trends are observable in the overall oxidation state of manganese. It is already known that global trends in manganese mineralization through geologic time were heavily influenced by the oxygenation of the atmosphere and oceans. In an extensive review of major deposits, Roy[43] explains that very few manganese deposits formed before the GOE, because there was no mechanism to oxidize the more soluble and mobile $Mn^{2+}$ ion into its less soluble counterparts, $Mn^{3+}$ and $Mn^{4+}$. Consequently, during the Archean, minor $Mn^{2+}$ residing in continental material was weathered and transported, accumulating in lakes and oceans. After the GOE, zones of oxygenated water in marine environments oxidized aqueous $Mn^{2+}$, enabling sedimentary deposits of Mn oxides and carbonates on a more widespread scale.[43] Although these broad, qualitative trends can be observed in Mn deposits in the geologic record, there has been no attempt to quantify changes in the oxidation state of Mn or any other redox-sensitive metal. We therefore also used the MED to quantify the average oxidation state of Mn across geologic time in order to assess the oxidation of the shallow crust.

In general, trends in average Mn oxidation state before ~600 Ma are less certain due to larger uncertainties caused by the

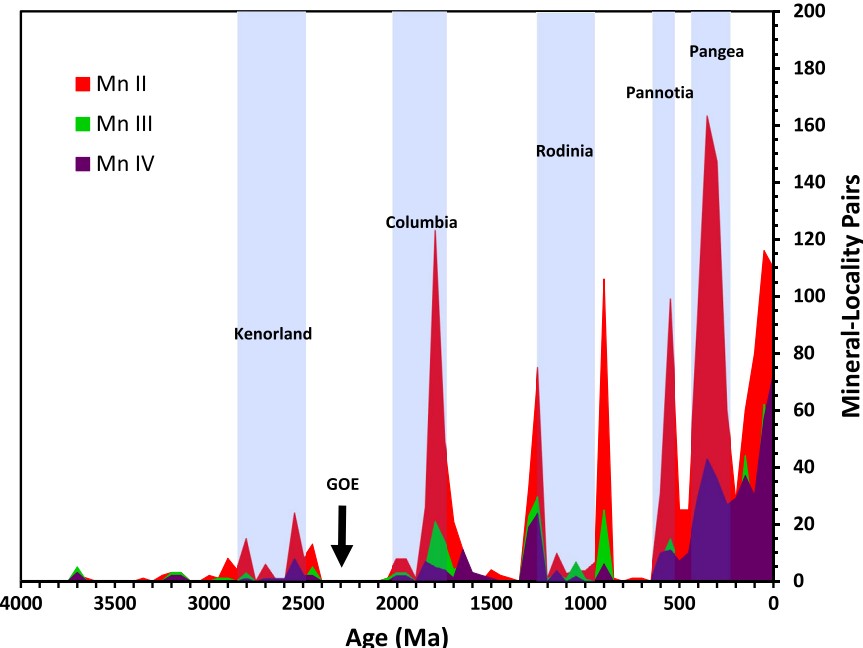

**Fig. 1 Mn mineral occurrence histogram.** A histogram showing the number of occurrences of mineral-locality pairs as a function of geologic time for all dated Mn-bearing mineral species, with data parsed into 50 million-year bins. Colors indicate oxidation state of Mn; Red = Mn(II), Green = Mn(III), Purple = Mn(IV). Blue bands represent intervals in which Earth's major landmasses are thought to have consolidated into a supercontinent.

smaller number of preserved deposits. An exception to this uniformly low oxidation state is an increase from +2 to greater than +2.5 in the average oxidation state of Mn in minerals that took place during the existence of Kenorland, from 3.0 to 2.5 Ga. This early oxidation of Mn predates most estimates of the timing of the GOE[4,14,18], making it initially appear as a curious anomaly. However, Johnson et al.[44] found evidence of extensive Mn oxidation in the absence of photosynthetic oxygen in Mn deposits of the Koegas Subgroup of South Africa (dated to 2.415 Ga)[45], pointing to a possible precursor of photosystem-II capable of biologically oxidizing $Mn^{2+}$. If this metabolic pathway was widespread before 2.4 Ga, it may explain this early, partial oxidation of Mn observed before the GOE. It is also possible that the hypothesized "whiffs" of atmospheric oxygen predating the GOE[8,22,23] were able to oxidize Mn in local environments and that these more highly oxidized Mn minerals were preferentially preserved, as suggested by the small number of localities in Earth's earliest history. This same preservation bias (which favors more oxidized Mn mineral occurrences) may have also shifted the post-GOE Proterozoic Mn record towards higher oxidation states that more closely resemble those of the Phanerozoic, dampening the sensitivity of our proxy to changes in atmospheric $O_2$ in the deep past.

After the GOE, a long-term increase in the proportion of Mn minerals containing trivalent and tetravalent Mn is shown in our dataset. This trend is especially visible in the proportions of +2, +3, and +4 oxidation states of Mn during the most recent 600 Ma, which shows a clear decrease in the occurrence of $Mn^{2+}$ mineral species relative to the occurrence of $Mn^{3+}$ and $Mn^{4+}$ species as a function of time (Fig. 2). Note that at any given time interval, preserved Mn minerals reflect a range of formation depths, and thus of redox states. In Earth's early history, Mn minerals would have formed under reducing conditions, whereas highly oxidized conditions are presumed to be more common in recent periods. Specifically, an overall increase in oxidation state is clearly visible from 600 Ma to the present. Although the average Mn oxidation state goes through minima and maxima during the Phanerozoic, these changes span multiple data points and are

larger than the uncertainties of the data (owing to the larger number of mineral-locality counts from recent occurrences). This result instills confidence that, even though errors may occur in individual mineral-locality data pairs, the very large amount of data overwhelms these random errors and reveals a steady increase in the average oxidation state of crustal Mn during the Phanerozoic.

The observed increase in Mn oxidation state can be directly compared to reconstructions of atmospheric oxygen (Fig. 3). Reconstructions by Berner and Canfield[15], Bergman et al.[16], and Holland[17] all contain local maxima in atmospheric oxygen content near 550–600, 250–300, and 50–100 Ma, and local minima near 400–450 Ma, 150–200 Ma, and the present day (Fig. 3A). Remarkably, the pattern of increases and decreases in atmospheric oxygen closely mirrors our data for the average Mn oxidation state, except that a time lag exists between a change in atmospheric oxygen and the corresponding change in Mn oxidation state.

To quantify this time lag, we modeled our Mn oxidation states and the atmospheric oxygen reconstruction of Bergman et al.[16] by fitting both data sets to a sinusoidal function that included fitting parameters for the period of oscillation, and a phase shift (representing the time lag). This fitting procedure yielded an oscillatory period of 35.7 Ma for the oxygen reconstruction versus a 35.0 Ma oscillatory period for the average oxidation state of Mn. The close agreement between these two values suggests that atmospheric oxygen concentration and Mn oxidation states changed in a correlated fashion over the Phanerozoic. The difference in phase shift between the two data sets was calculated to be ~66 (±1) Ma, indicating that changes in Mn oxidation state follow corresponding changes in atmospheric $O_2$ after an average 66 Ma time lag.

When this time lag is accounted for by shifting the oxygen reconstructions forward on the time axis by 66 Ma, the close correlation between the two variables is evident (Fig. 3B). This strong correlation suggests that changes in atmospheric oxygen content directly affect the oxidation state of manganese in Earth's shallow crust. However, the change occurs only on geologic time

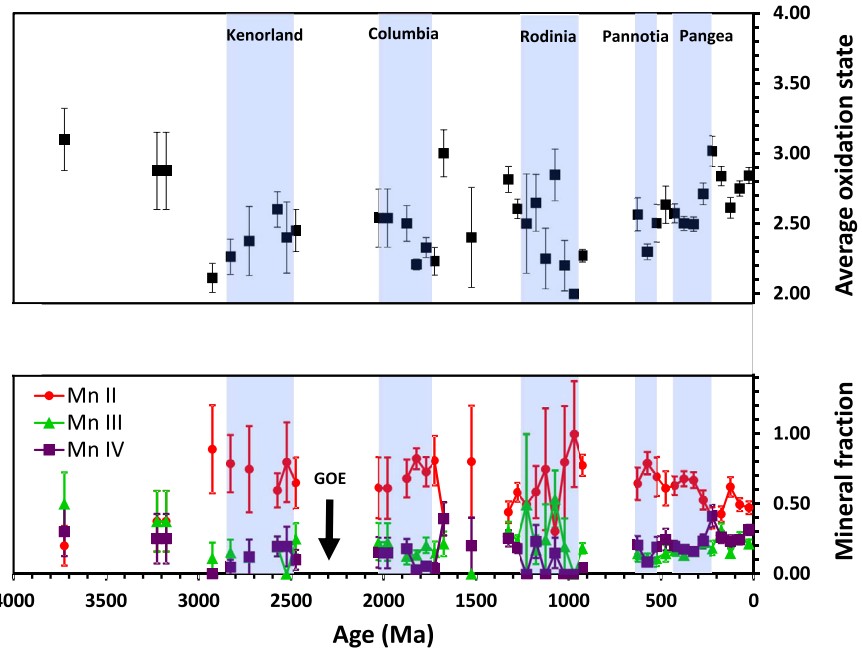

**Fig. 2 Individual and average Mn oxidation states as a function of geologic time.** Graph of fraction of Mn mineral occurrences (lower) and average Mn oxidation state (upper) as a function of geologic time, with data parsed into 50 million-year bins. Colored symbols indicate oxidation state of Mn; Red circles = Mn(II), Green triangles = Mn(III), Purple squares = Mn(IV). Blue bands represent intervals in which Earth's major landmasses are thought to have consolidated into a supercontinent. Uncertainties for oxidation state fractions are standard deviations calculated as the square root of the number of counts for an oxidation state divided by the total counts for all oxidation states in a given time bin, using an approximation from the negative binomial distribution as described in "Methods".

scales, and may proceed at varying rates depending on stage within a supercontinent cycle. Note that this 66-million-year delay contrasts with the hundreds of millions of years delay suggested by the rhenium content of molybdenite[35], which typically forms at greater crustal depth than manganese minerals. These differences suggest a possible strategy to use different mineral groups for documenting Earth's changing crustal oxidation state as a function of depth.

The time lag we observe between our Mn data set and atmospheric $O_2$ should be expected since the great majority of crustal material is not directly exposed to the atmosphere, and therefore cannot be immediately oxidized. In particular, manganese minerals typically form in the upper 1 km of the crust where sedimentary and supergene processes take place[46]. Thus, we interpret this lag as the average time necessary for Earth's shallow crust to equilibrate to a new atmospheric oxygen fugacity through a combination of: 1) oxidative weathering of near-surface material, 2) brittle fracturing of near-surface rock that exposes additional crustal material to meteoric waters and the atmosphere, and 3) tectonic processes such as subduction, volcanism, faulting, and orogeny that mix already oxidized surface material with deeper crustal and mantle reservoirs. During the Phanerozoic Eon, these processes have gradually oxidized the crust to greater and greater depth, thus equilibrating successively deeper minerals to the rising oxygen fugacity of the atmosphere. Manganese minerals, which typically form in the shallow crust, thus require on average a ~66 Myr time lag to allow sufficient mixing to take place. We should clarify here that no individual Mn mineral would have equilibrated to an atmosphere that existed 66 Myr in the past. Instead, we propose that over tens of millions of years, successively greater portions of Earth's crust were exposed to an atmosphere with steadily rising oxygen concentration. This process created a spatially averaged aggregate record of crustal Mn minerals that was always forced in the direction of change of atmospheric oxygen concentration, but with a time lag due to the slow mixing of crustal material.

This study is among the first to illuminate the timing and magnitude of the oxidation of Earth's crust over a substantial portion of Earth's history. Redox-sensitive metals record the redox state of large Earth reservoirs and provide an important and rarely explored class of proxies for oxygen fugacity through deep time. In particular, we have demonstrated aspects of the timing of the oxidation of Earth's shallow crust. We propose that analyzing large mineralogical databases of other abundant redox-sensitive metals, such as the other first-row transition metals, will provide new constraints on the oxygen fugacity of reservoirs such as the crust and ocean. Furthermore, analysis of different mineral groups may enable us to determine the timing of oxidation for different depths within the crust. These constraints will help bring into sharper focus a picture of the mechanisms by which Earth and life coevolved.

## Methods

Mineral-locality data pairs (i.e., a single mineral species occurring at a single locality) representing occurrences of Mn mineral species were harvested from the crowd-sourced mindat.org (as of 20 November 2015). The data set contained 22,064 Mn mineral-locality pairs, 2666 of which had associated geologic ages in the Mineral Evolution Database (rruff.info/ima).

All 560 known Mn mineral species were placed in three mineral lists according to the oxidation state of the Mn, one list each for the +2, +3, and +4 oxidation states. In many cases, the oxidation state could be uniquely determined by the charge balance of the nominal chemical formula. However, in some cases, the determination of Mn oxidation state was not straightforward. For sulfides, many of which contain significant anion-anion or cation-cation bonding, literature on the crystal structure was consulted to determine oxidation state. For minerals with multiple possible oxidation states, published spectroscopic data, and structural refinements were used to ascertain the oxidation states present. There is a small amount of overlap between lists, because a limited number of minerals contain Mn in multiple oxidation states. For example, the Mn analogue of magnetite, hausmannite ($Mn^{2+}Mn^{3+}_2O_4$), was included in both the +2 list and the +3 list.

To create the histogram of Mn mineral occurrence through time separated by oxidation state, all occurrences among the 2666 dated mineral-locality pairs of minerals on the +2, +3, and +4 lists were binned by age in 50 Myr bins and the frequencies of these occurrences were plotted on the vertical axis (Fig. 1). The mineral fraction of each oxidation state within a given age bin was calculated as the

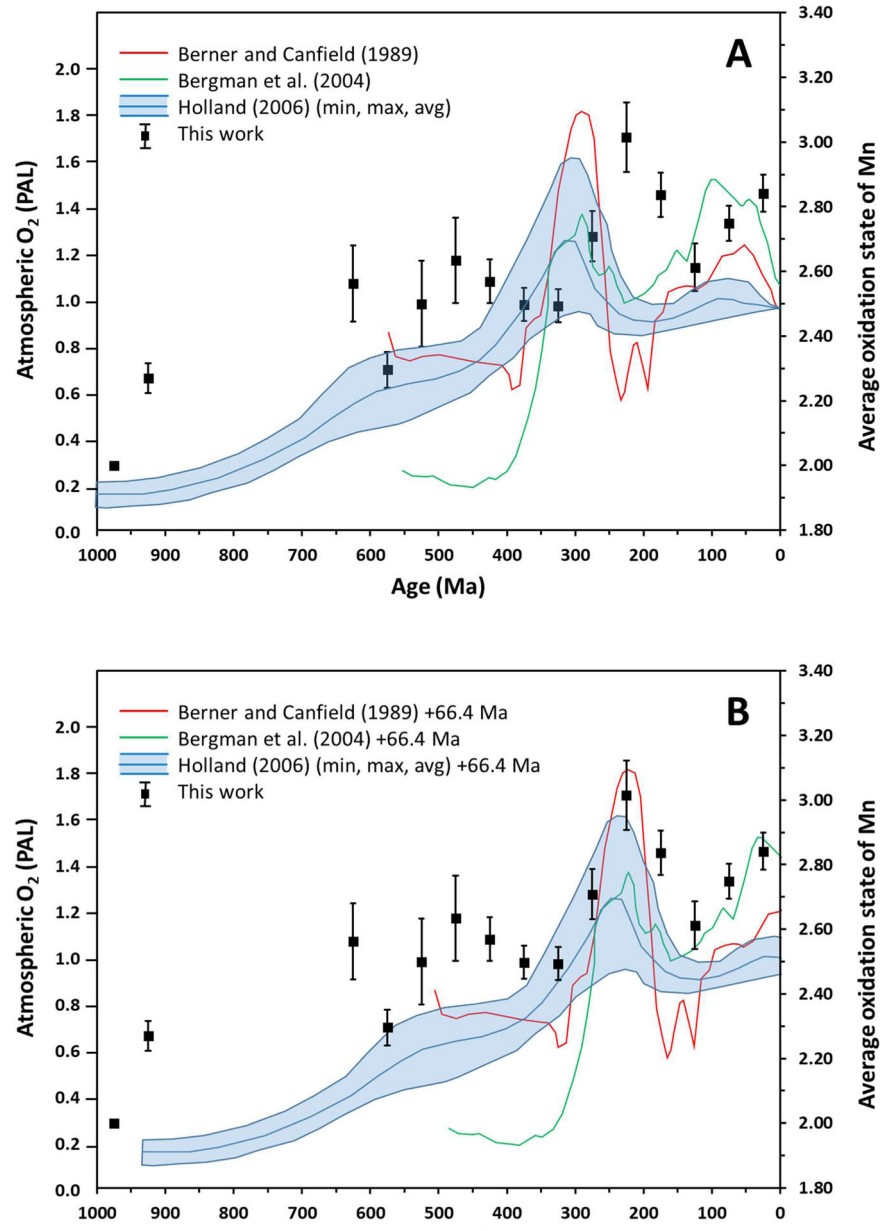

**Fig. 3 Comparison of average Mn oxidation state with atmospheric O₂ reconstructions. A** Graph of the average oxidation state of Mn according to Mn mineral occurrences (black squares, right vertical axis) over the most recent 1 billion years, overlain with reconstructions of atmospheric oxygen (left vertical axis) by Berner and Canfield[14] (red line), Bergman et al.[15] (green line), and the O₂ concentration range including minimum, average, and maximum by Holland[16] (blue lines). Uncertainties regarding the Mn oxidation state data points are standard deviations calculated (assuming a Poisson distribution of mineral-locality counts) as the square root of the sums of the variances of counts for the individual oxidation states according to propagation of errors. **B** Graph of the average oxidation state of Mn according to Mn mineral occurrences (black squares) over the most recent 1 billion years, overlain with reconstructions of atmospheric oxygen by Berner and Canfield[14] (red line), Bergman et al.[15] (green line), and the O₂ concentration range including minimum, average, and maximum by Holland[16] (blue lines), but with oxygen reconstructions shifted forward in time by 66.4 Myr. Note that when this time lag is accounted for, the average oxidation state of Mn in the geologic record displays a trend of minima and maxima that coincide closely with atmospheric oxygen concentrations.

number of occurrences of Mn minerals containing that oxidation state divided by the total number of Mn minerals occurring within that age bin (Fig. 2).

To determine errors, we fit the raw number of Mn mineral counts occurring within 50 Myr time bins (shown in Fig. 1) to various discrete probability distribution functions (Poisson, binomial, negative binomial, geometric, and hypergeometric). The fits were performed in R, and evaluated using the Bayesian information criterion (BIC), which allows comparison of fit across different distribution types. The best fit probability distribution function for our Mn mineral occurrences was found to be the negative binomial, with parameters of $size = 0.622$, $\mu = 11.569$, and $p = 0.051$. The standard error for the negative binomial distribution is $\sigma = \sqrt{N}/(1-p)$, where $N$ is the number of minerals of the given

oxidation state occurring within the time bin, and $p$ is the probability of Mn mineral occurrence per unit time. However, because $p$ is small for our dataset, $\sqrt{N}$ is a very good approximation for the standard error of a single measurement $N$. We therefore took the standard error for our raw Mn mineral counts to be $\sqrt{N}$, and errors on mineral fractions were calculated as $\sqrt{N}/T$, where $T$ is the total number of Mn minerals occurring within the time bin.

The average oxidation state of Mn for each time bin was calculated as $2x + 3y + 4z$, where the weighting factors $x$, $y$, and $z$ are the mineral fractions in the relevant time bin for the +2, +3, and +4 oxidation states, respectively (Fig. 2). Errors on the average oxidation state within a time bin were computed as the square root of the variance among all the individual occurrences of each oxidation

state within the time bin, according to standard propagation of error. Mineral fractions and average oxidation states were not computed for time bins for which there were five or fewer Mn mineral-locality pairs, since there were not enough data to obtain a statistically meaningful result.

To quantify the relationship between the Phanerozoic Mn oxidation states and the atmospheric $O_2$ reconstruction of Bergman et al.[16], each data set was fit to the same sinusoidal function. The function utilized was:

$$y = A \sin(\omega_a t - \varphi_a) + B \cos(\omega_b t - \varphi_b) \tag{1}$$

In this function, $y$ is the dependent variable of interest (either atmospheric $O_2$ concentration or average Mn oxidation state), $t$ is time, $\omega$ is frequency, $\varphi$ is phase shift, and $A$ and $B$ are scaling constants. The sine term was used to fit the broad, low-frequency background, and the cosine term was used to fit the higher frequency oscillations within each data set. The fitting was performed with an iterative, least-squares refinement procedure that minimized the sum of square errors between the data points and the calculated $y$ values at each available time point. This fitting procedure yielded oscillation periods of $2\pi/\omega_b(O_2) = 35.7$ Myr and $2\pi/\omega_b(Mn) = 35.0$ Myr, and a time lag between the oxygen and manganese data of $[\varphi_b(O_2)/\omega_b(O_2)] - [\varphi_b(Mn)/\omega_b(Mn)] = 66.4$ Myr. (Note that the phase shifts, $\varphi$, start out unitless and must be divided by frequency to be converted to units of time). For Fig. 3B, oxygen reconstructions from refs. [14–16] were moved forward in time by 66 Myr since this was the time lag that minimized the error between the Bergman et al.[16] oxygen reconstruction data and Mn oxidation state data. Similar fitting was not done for other oxygen reconstructions since these data sets were not available.

## Data availability
The raw data used in this study is freely available from the Mineral Evolution Database (MED) at http://rruff.info/ima/ by clicking "Mn", and then clicking "Export to Evolution". There are no restrictions on data availability. Each figure contains graphical representations of data that has been calculated or tabulated from the raw data in the manner described in the text.

## Code availability
No custom computer code was used in the preparation of this manuscript.

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

## Acknowledgements

This work was supported in part by the W. M. Keck Foundation (through the grant "The Co-evolution of the Geo- and Biospheres: An Integrated Program for Data-Driven Abductive Discovery in the Earth Sciences"), the Alfred P. Sloan Foundation, the Deep Carbon Observatory, the NASA Astrobiology Institute (ENIGMA team), a private foundation, and the Earth and Planets Laboratory at the Carnegie Institution for Science. Any opinions, findings, or recommendations expressed herein are those of the authors and do not necessarily reflect the views of the National Aeronautics and Space Administration.

## Author contributions

Data was assembled and formatted by J.J.G., J.R., and R.T.D. Data analysis was performed by D.R.H. with minor contributions by J.J.G., A.E. and G.H. Data was interpreted by D.R.H. with contributions by R.M.H., C.L., S.M.M., M.B.M., and R.T.D. The manuscript was written by D.R.H. with major contributions from R.M.H. and R.T.D.

## Competing interests

The authors declare no competing interests.
