## [Peer Review File · Nature Communications]

The oxidation of Earth's crust: Evidence from the evolution of manganese mineralsReviewers' Comments:

Reviewer #1:

Remarks to the Author:

Hummer and co-authors investigate the evolution of the crust's oxidation through time. They use the Mn oxidation state in minerals containing Mn in their formula to evaluate the redox state of the crust. They propose that the Earth's crust oxidizes through time and reflects the evolution of the atmosphere. They show that the oxidation occurs in the crust 66 Ma after changes in atmospheric oxidation.

The concept of using the different oxidation state of Mn minerals is clear, well explained and appears promising. This idea is novel and appealing. Links between the geosphere and the biosphere are poorly studied, especially in this way, where it is the atmosphere that influences the lithosphere and, we can imagine, ultimately the deep Earth. This makes the approach of Hummer and colleagues exciting and should be published in Nature Communications.

However, I have some concerns and comments that should be addressed.

Firstly, from my point of view, the major result is the coincidence between evolution pattern of atmospheric O₂ and crustal oxidation (through the evolution of Mn minerals valence state) during the Phanerozoic. But the period older than 350 Ma seems to require some explanations to assess the differences between Phanerozoic curves of atmospheric O₂ from Bergman et al. (2004), Berner and Canfield (1989) and from the Holland (2006) as well as the differences with crust oxidation evolution curves from Mn minerals. Indeed, the oxidation state during the period between 600 Ma and 350 Ma are differently described: low for Holland (2006), moderate for Bergman et al. (2004) and relatively high from this paper's data.

Secondly, in its first part, the authors discuss the evolution before 600 Ma at length. This part appears less convincing since variations on figure 2 do not clearly reflect an increase after the GOE nor after Neoproterozoic oxidation event. In my opinion, before 600 Ma, these values stay more or less the same within error bars. Nevertheless, the redox state evolution of the crust during Kenorland supercontinent appears clearly on the figure 2, but raises some questions about the representativeness of Precambrian data. Indeed, if the robustness of the analysis is clear for the recent period due to the number of data and the various localities, the paper will benefit to state on the number of data and the associated localities for the oldest periods as well as the formation depth. Are the localities in the different 50 Ma windows from different cratons, different area? Could the ratio between Mn²⁺, Mn³⁺ and Mn⁴⁺ be dependent of the sampling (or report)?

In the same way, the evolution during Rodinia is not discussed. It seems to exist more data during this interval and they seem to show a decrease during this period.

Despite these concerns, this subject, meaning the co-evolution of the oxidation of both crust and atmosphere is evidently a major topic on the Geosciences community.

I detailed minor comments below:

L. 120-121 « Note that at any given time interval, preserved Mn minerals reflect a range of formation depths, and thus of redox states ». The sampling of different depths should be more discussed, especially for the oldest data.

L.67. « Dramatic increase ». The term is a little strong. It is the last major step but minor compared to the GOE. Then again, why the oxidation of the crust would be more sensible to minor changes (i.e. after 600 Ma and following fluctuations) than to major changes (GOE) ?

Melina Macouin

Reviewer #2:

Remarks to the Author:

I have read the submitted manuscript, "The oxidation state of Earth's Crust: Evidence from the evolution of manganese minerals" by Daniel Hummer and co-authors. I have some comments on the manuscript that are described below.

Hummer et al. assess the average oxidation state of manganese in manganese minerals through geologic time, using a recent query of the MinDat database. They find that the average oxidation state of manganese may have increased from the Great Oxidation Event (~2.5 Ga) to present day as the result of the rise of oxygen at Earth's surface. They observe a ~66 my time lag between fluctuations in the average oxidation state of manganese minerals and hypothesized fluctuations in the pO₂ of Earth's atmosphere, and interpret this time lag to be representative of the time needed to oxidize manganese in Earth's crust via water infiltration and chemical weathering processes.

The manuscript is well written and the conclusions are interesting. The topic is very relevant for the broad readership of Nature Communications that seek to quantify the relationship between the solid Earth and the rise of oxygen, and the implications of the rise of oxygen on the petrological and mineralogical evolution of Earth materials. I enjoyed the manuscript, however in its present form, I am not certain that this manuscript presents a previously unknown observation, or adds significantly to the current understanding of oxidative manganese weathering and/or the formation conditions of manganese deposits on Earth. I think there is potential for the authors to expand on this idea and generate a truly new and novel constraint on the nature of the rise of oxygen in Earth's surface environments.

Here is a summary of my specific comments on the manuscript, which I hope illuminate my concerns.

1. First, I direct the authors to several articles in the Geological Society Special Publication No. 199, "Manganese Mineralization: Geochemistry and Mineralogy of Terrestrial and Marine Deposits", particularly contributions in that volume by Supriya Roy, G.P. Glasby, and Lev Gramm-Osipov. This volume illustrates nicely an understanding of manganese mineralization, where it is acknowledged that the rise of oxygen in Earth's atmospheres, surface waters, and oceans played a critical role in producing manganese ore bodies, and that the production of various types of manganese ores has changed as pO₂ at Earth's surface has varied. If Figures 1 and 2 in this submission present a new understanding than those summarized by Supriya Roy in the second article of the volume I mention here, I am not sure it is emphasized sufficiently in this submission.

2. This submission could be made stronger by considering the various formation mechanisms of the lithological units from which the manganese mineral occurrences are derived, and how these mechanisms might relate to atmosphere or ocean oxygen levels. Are the observations in Figure 1 mostly manganese ore bodies? Which are land-based deposits, and which are marine-based (i.e., which were formed subaerially and which were formed in submarine environments)? The supergene-type mechanisms that the authors describe in this submission seem most relevant for land-based manganese mineralization. See the article in the volume I mention above by G.P. Glasby, Figure 1, for an interesting observation – that land-based manganese ore bodies peak in formation just after the Great Oxidation Event, but do not appear to form in such abundance again through most of the Proterozoic and Phanerozoic. Some manganese ore bodies are marine deposits, which depend on high pO₂ in Earth's atmosphere and surface waters, but do not conform to the model described in this submission, where manganese minerals are oxidized in situ via supergene-type processes. Why then, the offset from atmospheric pO₂ constraints, if some of these occurrences describe marine depositional environments? If a significant portion of the manganese minerals in this submission are from marine-based ore deposits, proximity to anoxic waters can be important in their formation, and perhaps the authors should consider the relevance of ocean anoxic events in their story. Can the

authors demonstrate that land-based deposits conform to the in situ supergene model they propose in this submission, and use the marine-based deposits to comment on the widespread nature (or not) of oxic/anoxic waters through time? This could potentially transform this contribution from a concise visualization of manganese mobility and mineralization, into an entirely new constraint on the global nature (or not) of the rise of oxygen in the atmospheres, surface waters, and oceans through the Proterozoic and Phanerozoic.

3. These are minor suggestions/comments that reflect my reading of the manuscript as a "first-time reader" for the authors to consider:

a. There are no line numbers on the manuscript I downloaded, but I have added them in Word and reference them here. Citations for the following statements should be added:

Line 42 "...most influential processes in Earth's 4.5 billion year history."

Line 43: "...free O₂ directly enable the evolution of modern animals such as humans,"

Line 44: "...changed Earth's...mineralogy, petrology, and geochemical cycles."

Line 58: "...rather than a single event,"

Line 60: "...must have taken place before the GOE,"

b. None of the figures reproduce well in gray scale. It is particularly difficult to distinguish between the various Mn oxidation state data series.

c. At lines 115 and 118, I recommend against using "clear" to describe the trends in Figures 1 and 2. I see the change, but whether it is "clear" or "subtle" may be subject to the reader's opinion. Related, in Figure 2a, what are the high values in average oxidation state that appear prior to 3 Ga, and just after the box marked "Columbia"? In Figure 2b, why does the y-axis scale extend to >1?

Reviewer #3:

Remarks to the Author:

The authors present a new Mn-mineral-based proxy for the redox state of the upper continental crust through time, with a focus on the last 1 billion years of Earth history, and compare their proxy results to published reconstructions of atmospheric O₂ levels. It is suggested that the occurrence and average Mn valence state of manganese minerals increases in step with an increasing redox state of the shallow crust, which in turn qualitatively follows atmospheric O₂ with an estimated time lag of 66 Ma. This lag is interpreted as the timescale for the response of the shallow crust to changes in surface O₂ levels. Proxy data are provided by the mindat.org database of mineral occurrences and localities. This is presented as one of the first uses of a large mineralogical databases in evaluating Earth's geochemical evolution, and it is suggested similar approaches with this and other mineral proxies may be used to further address the redox evolution of the crust and other Earth reservoirs.

I do not recommend this article for publication in Nature Communications. The authors demonstrate that there is some intrinsic merit in the use of mineral databases for tackling big geochemical questions, however in its current form this contribution is insufficiently quantitative, and processes used to explain the observed time lags are underdeveloped and thus the argument that any time lag should exist appears rather circumstantial. Enough details are left vague in what this newly proposed proxy actually records (and how), that I cannot agree with the author's claims that it will be a novel and powerful tool to move the field forward.

The use of "Mn minerals" as defined in the text is evidently practical given the nature of the available database, but a sentence is required justifying why only minerals in which Mn is a major element is used. What wt% of upper crustal manganese is present in these minerals rather than as a minor element or in amorphous phases, and are the results affected by their omission?

Numerous references are made to variations in crustal mineralogy and inferred redox state with depth, even within the upper 1 km, but plots of Mn mineral occurrence and/or redox state versus depth are not shown. If Mn valence state varies systematically with depth, what preservation biases could come into play if the age and depth of mineral deposits is in some way correlated? The terms "shallow crust" and "upper crust" are used interchangeably throughout the text without the formation or occurrence depth of any Mn minerals being provided, which left this reviewer rather confused.

Could you instead be observing changes in redox environment with depth rather than a delayed crustal response to rising atmospheric pO₂? Without depth info being presented, the reader could be led to conclude as much.

Time lags – I have a few concerns here. First, apparent disagreement with the Re in molybdenite record of several hundreds of millions of years. The point about these minerals forming at greater depth than Mn minerals needs expansion and clarification – there is manganese in the crust at all depths so in what species does it reside in these greater depths and what does the redox state of deeper crustal manganese indicate? Again, a lack of clarity about Mn mineral variability with depth, as I mentioned above, is a big hindrance to understanding how these proxies should work. For example, if redox changes creep deeper into the crust over time, then by necessity a group of minerals with the same apparent age but different depths are actually reflecting equilibration with different atmospheric redox states?

Related, is the issue of the time lag itself and the proposed mechanism. A 66 Ma time lag in the crustal Mn mineral response to changes in atmospheric O₂ should rule out certain mechanisms. For example, 1) oxidative weathering, and 2) brittle fracturing and exposure of the crust to the hydrosphere and atmosphere are both processes that may take millions of years, however the final alteration products (Mn minerals here) will not be equilibrated solely with the atmospheric composition which existed when said weathering or fracturing initiated (i.e. 66 Ma before the completion of mineralization), but reflect a time averaged interaction over the entire alteration period. Put another way, the authors must clearly explain their seeming argument that an Mn mineral formed at time T forms in equilibrium with the atmosphere that existed at time T minus 66 million years and not its ambient geochemical environment. Again I must stress that the data presented in this argument are interesting and potentially powerful but let down by a lack of detail and any attempt at a quantitative treatment of the processes which underpin how the proxy and its time lag works. The mere fact that this time lag was not predicted a priori, but instead comes out as a result of comparison to atmospheric reconstructions begs the question how Mn minerals can be used as an independent redox proxy through time.

Line by line comments on the text below. Generally, very well written.

23: Suggest this statement needs qualification. There are methods to measure crustal redox.

33: Use this definition for shallow crust significantly throughout and avoid flip-flopping between this and 'upper' crust.

41: Be specific. Cyanobacteria or oxygenic photosynthesizing.

43: Suggest replacing 'modern animals such as humans' for something less loaded, perhaps just 'animals'.

48: Do none of these reconstructions extend back farther than 2.5 Ga? I believe Lyons et al. also provide an O₂ reconstruction for the Archaean.

53: Replace upper crust with shallow crust for consistency, or define the separate terms clearly.

55: Cyanobacteria.

59: Oxygenic photosynthesis.

64: 'negligible' is vague. Provide published numerical estimate in PAL.

75: Did all of these transition metals really undergo redox changes at the GOE? I believe Ni remains 2+ throughout.

77: Provide a clear definition of how redox state is being defined in the context of this study.

81: There are other processes to also consider in the redox state of the mantle.

84: How did this Re proxy work, exactly?

94: Are the minerals themselves in this database dated, or just the host rocks? What % of Mn in the shallow crust is tied up in stoichiometric Mn minerals?

102: Define 'primary mineralization' and relevant mechanisms.

109: There is hardly a consensus on this Johnson model of late evolution of oxygenic photosynthesis, and this mechanism can only explain surface oxidation of Mn, not within the top 1 km of crust.

120-121: Why are we not shown any plots for the this range of formation depths vs. age or Mn valence?

156: I will reiterate my point about a time lag requiring further process-based explanation with regards to minerals forming in equilibrium with their ambient redox conditions which are unlikely to

reflect 66 Myr-old atmospheres at the time of mineralization.

163: This subduction explanation is the only one for which mineralization could occur in an environment out of communication with the atmosphere for many Myr.

Below, we list reviewer comments individually and provide a response, also noting any changes we made to the manuscript in order to address the reviewer's concerns. All revisions to the manuscript are denoted with red font, and all responses to reviewer comments below are denoted in blue font. We are happy for these responses to be shown to reviewers should the editor deem it useful.

Reviewer 1:

1) But the period older than 350 Ma seems to require some explanations to assess the differences between Phanerozoic curves of atmospheric O₂ from Bergman et al. (2004), Berner and Canfield (1989) and from the Holland (2006) as well as the differences with crust oxidation evolution curves from Mn minerals.

Author response: It makes sense that the Mn oxidation data points do not perfectly overlap with the atmospheric oxygen reconstructions, since these values are in different units, and although they are certainly correlated there is a complex and nonlinear conversion between the two. As for the oxygen reconstructions, we are uncertain why the Bergman et al. reconstruction differs substantially from the other two reconstructions prior to 350 Ma. However, for purposes of comparing these trends, the absolute values are much less important than the direction of the trends. For the time span of 500-350 Ma, all three oxygen reconstructions and the Mn data from our study show a steady to slightly decreasing trend in redox state over that time period. The direction of the trend, and therefore the direction of forcing, remains consistent for the entire time period shown in the graph, which supports our main conclusion that atmospheric O₂ is driving Mn oxidation in the crust.

2) Secondly, in its first part, the authors discuss the evolution before 600 Ma at length. This part appears less convincing since variations on figure 2 do not clearly reflect an increase after the GOE nor after Neoproterozoic oxidation event. In my opinion, before 600 Ma, these values stay more or less the same within error bars. Nevertheless, the redox state evolution of the crust during Kenorland supercontinent appears clearly on the figure 2, but raises some questions about the representativeness of Precambrian data.

Author response: We completely agree that trends seem vague and inconsistent in the time period prior to 600 Ma. We believe the reason is that the number of available Mn occurrences in our datasets begins to rapidly drop as age increases beyond the 600 Ma threshold, and as this happens, the statistics become less precise, the error bars increase, and the data become increasingly susceptible to preservation bias. Thus, in the manuscript we restrict our discussion of the Precambrian data to potential explanations for the unusually high data points before 3.0 Ga, and avoid discussion of the vague trends (which may or may not be real) during the formation of early supercontinents. Of these, the apparently increasing trend during Kenorland is of the most potential interest since it runs counter to the expected trend from preservation bias. However, we don't have enough confidence in the limited and potentially biased data from this time period to make concrete assertions about what was happening.

3) In the same way, the evolution during Rodinia is not discussed. It seems to exist more data during this interval and they seem to show a decrease during this period.

Author response: We believe that unusual tectonics and geochemistry is happening during the existence of Rodinia. Specifically, one of the coauthors of this paper, Chao Liu, has published an entire paper on the geochemistry of Rodinia, and suggests that an abundance of non-arc magmatism produced mineralogy that was dominant in high field strength lithophile elements such as Zr and Nb, but eroded away deposits of lower field strength, higher solubility elements such as Mn. We have added text on lines 105-110 in the revised manuscript (as well as the Liu et al. reference) to address this question.

4) The sampling of different depths should be more discussed, especially for the oldest data.

Author response: We completely agree with the reviewer that an examination of the formation depths of the Mn mineral occurrences would greatly assist in interpreting the trends in our data, and revealing further detail about how oxidation of the crust occurred. However, there is currently no formation depth information in the Mineral Evolution Database, and our data set consisted of 2666 individual mineral-locality data pairs from around the world and across geologic time. This means that any unbiased examination of formation depth requires us to go to the primary literature by hand, and enter average formation depths for all 2666 mineral-locality pairs, which will take many years of work. We are actually in the process of working out ways of adding formation depth, pressure, temperature, and other crucial data to our database, but because results on this front are potentially years away, we are unfortunately unable to comment on trends in formation depth at this time.

5) The term (dramatic increase) is a little strong. It is the last major step but minor compared to the GOE. Then again, why the oxidation of the crust would be more sensible to minor changes (i.e. after 600 Ma and following fluctuations) than to major changes (GOE)?

Author response: We understand this comment, and we have changed the word “dramatic” to “notable” in order to not overstate the magnitude of this increase (line 68 of the revised manuscript). As for why Mn oxidation state would be more sensitive to changes after 600 Ma than to changes during the GOE – we don’t believe that it actually is. As we note in Lines 133-136 (of the revised manuscript), we suspect that as age increases, less oxidized Mn deposits are more susceptible to being weathered away, preferentially leaving behind the more oxidized deposits. This preservation bias would therefore skew older data towards higher oxidation states, and dampen the apparent sensitivity of Mn oxidation state to atmospheric oxygen.

Reviewer 2:

1) I am not certain that this manuscript presents a previously unknown observation, or adds significantly to the current understanding of oxidative manganese weathering and/or the formation conditions of manganese deposits on Earth.

Author response: The goal of this study was not to investigate the specific mechanisms by which oxidizing conditions create Mn mineral deposits, which we agree are reasonably well understood. The goal was instead to use large data resources to quantify *how prevalent* those oxidizing conditions were in Earth's crust across geologic time. To our knowledge, such a quantification has never been done before. So, we argue that we have a three-fold contribution to new knowledge and understanding of Earth's redox history: A) We have demonstrated that an entirely new type of proxy for redox conditions in the crust correlates with proxies for redox conditions in the atmosphere, B) Our technique can be applied to other mineral systems to place new constraints on redox conditions in various Earth reservoirs, therefore making it applicable to answering multiple important questions in Earth system science, and C) We have documented for the first time the timing of oxidation of the crust vs. oxidation of the atmosphere. We contend that these are all important and novel contributions to our understanding of the coevolution of the geosphere and biosphere.

2) First, I direct the authors to several articles in the Geological Society Special Publication No. 199, If Figures 1 and 2 in this submission present a new understanding than those summarized by Supriya Roy in the second article of the volume I mention here, I am not sure it is emphasized sufficiently in this submission.

Author response: First, the authors thank the reviewer for pointing us towards this summary. The first author actually looked through this very volume while preparing the manuscript, and somehow missed the important summary by Roy (1997) regarding the way in which atmospheric and oceanic oxygenation changed global Mn mineralization. We have added a paragraph to our manuscript (starting at line 111 of the revised manuscript) to make it clear that these qualitative trends have been recognized previously. We do, however, also think that our quantification of the average oxidation state of Mn (which is the first of its kind) is an important new result with new implications that are described in the rest of the paper (see the response to point #1 above). We hope that this new text will adequately summarize previous observations and make our contribution more clear. Our thanks again to the reviewer for pointing out this oversight.

3) This submission could be made stronger by considering the various formation mechanisms of the lithological units from which the manganese mineral occurrences are derived, and how these mechanisms might relate to atmosphere or ocean oxygen levels..... This could potentially transform this contribution from a concise visualization of manganese mobility and mineralization, into an entirely new constraint on the global nature (or not) of the rise of oxygen in the atmospheres, surface waters, and oceans through the Proterozoic and Phanerozoic.

Author response: To answer the reviewer's question regarding how many terrestrial vs. marine deposits are represented in our data, and how these trends might be used to tease apart oxidation in various reservoirs, we completely agree that this would be a worthwhile and revealing exercise. However, we again need to stress that we are conducting this study using *extremely* large datasets that are not amenable to separating different classes of mineralization events. Our dataset represents 2666 individual mineral-locality data pairs, spanning all of the known terrestrial and marine modes of manganese mineralization for all of geologic history. Because the mode of

occurrence is not a specific field in the Mineral Evolution Database, conducting such a study and examining those trends would require going to the primary literature, documenting, and separating the modes of occurrence of all 2666 instances individually, which will take many years of effort. We therefore can't comment on trends indicated by modes of occurrence at this time. However, we are currently working on ways of incorporating paragenetic mode into more of our database, and in the meantime, we believe the trends available from the current dataset point to new insights and a new set of proxies that mineralogists, geochemists, and geochronologists can begin taking advantage of.

Also, we want to take the opportunity to clarify what we are proposing. Within this point, the reviewer mentions multiple times that there are modes of Mn mineralization that don't conform to the "supergene-type mechanism" that we propose. We're uncertain why the reviewer had the impression that we are proposing a supergene mechanism for all oxidized Mn deposits. When we talk about the oxidation of Mn in Earth's crust, we are speaking very broadly about geological processes that mix and cycle material between the atmosphere, hydrosphere, crust, and mantle over geologic timescales, which would serve to equilibrate the fO_2 of the crust to the fO_2 of the atmosphere. We do not in any way mean to restrict the mechanism of crustal Mn oxidation only to terrestrial supergene processes.

4) There are no line numbers on the manuscript I downloaded.

Author response: We apologize for the oversight, and have added line numbers to our revised manuscript.

5) Citations for the following statements should be added:

Line 42 "...most influential processes in Earth's 4.5 billion year history."

Line 43: "...free O₂ directly enable the evolution of modern animals such as humans,"

Line 44: "...changed Earth's...mineralogy, petrology, and geochemical cycles."

Line 58: "...rather than a single event,"

Line 60: "...must have taken place before the GOE,"

Author response: For the statement on line 58, we have cited reference 19 for the nature of the GOE, and for the statement on line 60, we have cited references 20 and 10 as the possible scenarios for the evolution of photosynthesis before the GOE. We therefore argue that sufficient citations (occurring within the sentence) are already present for those statements. The other three statements (i.e., the rise of oxygen was a highly influential process; free atmospheric O₂ enabled the evolution of animals; free O₂ changed geochemical cycles) are, to the authors, self-evident enough and supported by such a large amount of literature, that we did not view it as useful to cite individual papers to support them. We aren't aware of anyone in the relevant scientific communities that would disagree with these statements, so we request that the importance of the GOE be treated as a common understanding so that we can save our allotted citations (50 for a research article) for more specific claims.

6) None of the figures reproduce well in gray scale. It is particularly difficult to distinguish between the various Mn oxidation state data series.

Author response: We thank the reviewer for pointing out the difficulty. The figure in which this issue most comes into play is Figure 2, and we have therefore prepared a new version of Figure 2 in which the different oxidation states are not only colored differently, but are symbolized with different shapes (circle, triangle, and square). We hope this will assist readers who only have access to black and white printouts of the paper, although since Nature Communications is open access, we hope that nearly all readers will have access to the colored electronic copies.

7) At lines 115 and 118, I recommend against using “clear” to describe the trends in Figures 1 and 2. I see the change, but whether it is “clear” or “subtle” may be subject to the reader’s opinion.

Author response: We agree that the change in the entire period after the GOE may not be clear at first to everyone, so we have edited the language in this sentence on lines 137-138 in the revised manuscript.

8) Related, in Figure 2a, what are the high values in average oxidation state that appear prior to 3 Ga, and just after the box marked “Columbia”?

Author response: The very oldest data point is from Mn mineral occurrences from the Tabba Tabba pegmatite in Western Australia, and an ore body in Gogebic Co., Michigan, that are both dated to around 3.7 Ga. The other data points arise from more localities than it is practical to list here, although we are happy to make the raw data table available in the online supplements of our publication if it would be useful. As to why they show relatively high values of average Mn oxidation state, we speculate in our manuscript (lines 133-136 in the revised version) that the Archean examples represent the preferred preservation of the higher oxidation state (and less erodible) deposits. We are uncertain as to the reason for the post-Columbia data point, but we do caution in the manuscript that the larger error bars on older data points warrant a great deal of caution in overinterpreting data, especially from before the Phanerozoic.

9) In Figure 2b, why does the y-axis scale extend to >1?

Author response: We can understand the confusion with this since a fraction will never exceed 1. However, extending the y-axis was the only way to display the full extent of error bars on some data points, and to make room for a legend.

Reviewer 3:

1) The authors demonstrate that there is some intrinsic merit in the use of mineral databases for tackling big geochemical questions, however in its current form this contribution is insufficiently quantitative, and processes used to explain the observed time lags are underdeveloped and thus the

argument that any time lag should exist appears rather circumstantial. Enough details are left vague in what this newly proposed proxy actually records (and how), that I cannot agree with the author's claims that it will be a novel and powerful tool to move the field forward.

Author response: In our manuscript, we plot a histogram of mineral occurrences, mineral fractions of different oxidation states, and the average oxidation state, all as a function of geologic time. We conducted a statistical analysis of our data, finding that it fits best to a negative binomial distribution, used this to compute error bars for all data points, and used a statistical least squares fitting procedure, complete with analysis of errors, to compare the trend in oxidation state with previous proxies for atmospheric oxygen (see methods section). We are unable to think of any quantitative analysis of our dataset that was not conducted and presented in the manuscript, so we're unsure why this reviewer considered our study insufficiently quantitative. We would appreciate it if the reviewer could elaborate on what quantitative information he/she feels is missing.

As for explaining the time lag, we outline the kinds of processes we feel are responsible for the lag near the end of our manuscript. We are simply proposing that this proxy records the gradual oxidation of Earth's shallow crust over the Phanerozoic, but we feel that making statements any more detailed than these would require the sorts of comparisons with formation mode, formation depth, etc., recommended by Reviewers 1 and 2 that were not feasible with our datasets. We look forward to conducting these studies once this type of data becomes available, but in the meantime, we did not want to over-interpret or make unwarranted conclusions from the currently available data.

2) The use of "Mn minerals" as defined in the text is evidently practical given the nature of the available database, but a sentence is required justifying why only minerals in which Mn is a major element is used. What wt% of upper crustal manganese is present in these minerals rather than as a minor element or in amorphous phases, and are the results affected by their omission?

Author response: This is an entirely fair question, since Mn is known to replace Fe so easily as a minor element in nominally Fe-bearing mineral species. A calculation of the wt% of crustal Mn as a major element in nominal Mn minerals vs. as a minor or trace element in bulk crustal rock would require data on the volumes of numerous Mn mineral deposits that doesn't appear to be available in the literature. However, we can use a few key pieces of data to make some inferences. According to a USGS report on Mn, we know that Mn constitutes 0.16% of mafic rock and 0.06% of felsic rock in the crust, and that the overwhelming majority of this Mn occurs in the +2 oxidation state (oxidized Mn mineral species generally require a process that releases and oxidizes Mn²⁺ from the primary bulk rock). Therefore, we argue that adding this minor (but significant) source of Mn that is uniformly in the +2 oxidation state would most likely shift our dataset of average Mn oxidation states to slightly lower values, but is unlikely to effect the temporal trend seen in our dataset of major Mn minerals. We have added text on lines 94-98 of the revised manuscript plus the USGS reference to clarify this.

3) Numerous references are made to variations in crustal mineralogy and inferred redox state with depth, even within the upper 1 km, but plots of Mn mineral occurrence and/or redox state versus depth are not shown. Could you instead be observing changes in redox environment with depth rather than a delayed crustal response to rising atmospheric pO₂? Without depth info being presented, the reader could be led to conclude as much.

Author response: Here the reviewer proposes an intriguing alternative hypothesis. We know from the analysis of our dataset that average Mn oxidation fluctuates and rises during the Phanerozoic, and that this trend strongly correlates (once the time lag discussed in the manuscript is accounted for) with atmospheric O₂ proxies. Whether this rise in oxidation state also correlates with, and is partly due to, a shift in the average depth of Mn mineralization to shallower depths (which would favor increased oxidation state) during the Phanerozoic is a question that cannot be answered without formation depth data for most or all of the Mn mineral occurrences in our database. Unfortunately, as noted above under Reviewer 1 (Point 4), we do not currently have access to depth data and obtaining enough depth data to examine this relationship is not currently feasible (although we are actively working to make it more feasible as we learn ways of incorporating new data into the database). In the meantime, we believe the strong correlation with atmospheric O₂ proxies makes the delayed oxidation of the crust by the newly oxygenated atmosphere the most likely explanation using the available data.

4) Time lags – I have a few concerns here. First, apparent disagreement with the Re in molybdenite record of several hundreds of millions of years. The point about these minerals forming at greater depth than Mn minerals needs expansion and clarification – there is manganese in the crust at all depths so in what species does it reside in these greater depths and what does the redox state of deeper crustal manganese indicate? Again, a lack of clarity about Mn mineral variability with depth, as I mentioned above, is a big hindrance to understanding how these proxies should work. For example, if redox changes creep deeper into the crust over time, then by necessity a group of minerals with the same apparent age but different depths are actually reflecting equilibration with different atmospheric redox states?

Author response: First, we're not concerned that there is not complete agreement between the Re record and the Mn record. Since Re and Mn have redox transitions at different oxygen fugacities and, as noted in the manuscript, form at very different depths and geologic settings, it is not surprising that the time lags for Re oxidation vs. Mn oxidation are different. Qualitatively speaking, deeper minerals would be expected to take a longer time to equilibrate to a new atmospheric fO₂ than shallow minerals, which is the relationship observed in these two records. As for parsing the data by depth of formation, we completely agree with the reviewer that this would provide important insights. However, as noted in previous points, it is unfortunately not currently feasible to assemble the depth data that would be required to undertake that line of investigation.

5) Related, is the issue of the time lag itself and the proposed mechanism. Put another way, the authors must clearly explain their seeming argument that an Mn mineral formed at time T

forms in equilibrium with the atmosphere that existed at time T minus 66 million years and not in ambient geochemical environment.

Author response: The reviewer makes a good point here about the nature of the time lag that needs clarification. We are not necessarily arguing that a Mn mineral formed at time T forms in equilibrium with the atmosphere from time T minus 66 million years. Instead, we argue that successively greater portions of the shallow crust were exposed and allowed to equilibrate with the ambient atmosphere over a time window of tens of millions of years. This results in an aggregate record for the redox state of crustal Mn minerals that lags behind the trend in atmospheric oxygen, because at any given time, some crust has been exposed to recent atmosphere, while some has not. We have added text in lines 183 and 187-193 of the revised manuscript to address this point.

6) Again I must stress that the data presented in this argument are interesting and potentially powerful but let down by a lack of detail and any attempt at a quantitative treatment of the processes which underpin how the proxy and its time lag works. The mere fact that this time lag was not predicted a priori, but instead comes out as a result of comparison to atmospheric reconstructions begs the question how Mn minerals can be used as an independent redox proxy through time.

Author response: We completely understand the reviewer's point that the time lag warrants a more detailed explanation. However, since we are using a very large data set that broadly represents the Earth's shallow crust across geologic time, we feel that only broad interpretations are appropriate at this time. A more detailed investigation of the precise mechanisms responsible and their extent would be fascinating, and while we're working towards being able to do such studies, they would require additional data that are not currently accessible. We should stress that the Mineral Evolution Database used in this study is quite new and is still being updated, as we work to find ways to add the sorts of data fields that the reviewers have mentioned in an accurate and efficient way. In the meantime, we have done the most quantitative treatment we can of the Mn oxidation data we have access to, and tried to draw appropriate conclusions at an appropriate level of detail.

7) Next are series of short, line by line comments:

23: Suggest this statement needs qualification. There are methods to measure crustal redox.

Author response: We have edited this text to acknowledge that there are existing methods.

33: Use this definition for shallow crust significantly throughout and avoid flip-flopping between this and 'upper' crust.

Author response: We have changed our description in line 53 to be consistent with the previous definition.

41: Be specific. Cyanobacteria or oxygenic photosynthesizing.

Author response: We have made the change and specified cyanobacteria.

43: Suggest replacing ‘modern animals such as humans’ for something less loaded, perhaps just ‘animals’.

Author response: We have deleted the phrase “such as humans”.

48: Do none of these reconstructions extend back farther than 2.5 Ga? I believe Lyons et al. also provide an O₂ reconstruction for the Archaean.

Author response: Although the most reliable data are definitely for < 2.5 Ga, we have changed the claim to acknowledge data sets that go further back.

53: Replace upper crust with shallow crust for consistency, or define the separate terms clearly.

Author response: As noted, we made this replacement.

55: Cyanobacteria.

Author response: We have made the change.

59: Oxygenic photosynthesis.

Author response: We have added the clarification.

64: ‘negligible’ is vague. Provide published numerical estimate in PAL.

Author response: We have added a more precise figure with a reference.

75: Did all of these transition metals really undergo redox changes at the GOE? I believe Ni remains 2+ throughout.

Author response: We have deleted Ni and Cu to be on the safe side. Fe, Mn, and Co definitely underwent changes in redox state at some point during the GOE.

77: Provide a clear definition of how redox state is being defined in the context of this study.

Author response: By redox state, we mean the spatially averaged fO₂ of crustal material, as would be estimated by the combination of redox buffer minerals that coexist in various parts of the crust. We have added text on page 78 of the revised manuscript to clarify this.

81: There are other processes to also consider in the redox state of the mantle.

Author response: Our statement is not meant to exclude other possible processes that can affect the redox state of the mantle, we are simply mentioning and citing the mechanism that has been most extensively studied in the recent literature.

84: How did this Re proxy work, exactly?

Author response: Rhenium is a redox sensitive metal with low abundance and low oxidation state (primarily +2) in Earth's crust that would have required more oxidized conditions in order to mobilize into aqueous solutions. Re⁴⁺ substitutes for Mo⁴⁺ in molybdenite, which is primarily a hydrothermal mineral, and therefore higher concentrations of Re in molybdenite would correspond to greater Re mobility and therefore higher fO₂ conditions. We refer the reader to Golden et al. (Ref. 34) for more details on this proxy.

94: Are the minerals themselves in this database dated, or just the host rocks? What % of Mn in the shallow crust is tied up in stoichiometric Mn minerals?

Author response: In some cases the mineral itself is directly dated, in other cases the mineralization of a particular element is dated (in which case all minerals containing that element are assigned that age), and in other cases a primary host rock is dated (in which case all minerals occurring within the host rock are assigned that age). In a database as large and varied as the Mineral Evolution Database, it is difficult to keep everything perfectly consistent while still taking advantage of the totality of the data. However, we do add data points individually, and try to ensure their accuracy as much as possible. In the case of geologic ages, our usual rule is to include ages in which the mineral itself or a primary igneous or metamorphic host rock has been directly dated, and to exclude ages from sedimentary rocks, or situations in which the Mn-bearing phase is obviously a product of secondary weathering processes. These decisions are made on a case by case basis, and the MED always contains a note on how the age was obtained, and is coded to one of the situations outlined above. Adding data in this fashion requires a great deal of time and attention, which is why some data fields such as formation depth are not yet included in the MED.

As for the % of Mn that resides in nominal Mn mineral species, the data needed to calculate this doesn't appear to be available in the literature. However, in point #2 above we argue from USGS data that the small but significant amount of Mn that is present as a minor or trace element in other phases is uniformly in the +2 oxidation state, and is unlikely to affect the trends discussed in this manuscript. We have added text on lines 94-98 of the revised manuscript to address this concern.

102: Define 'primary mineralization' and relevant mechanisms.

Author response: By primary mineralization, we mean crystallization from either a melt or a hydrothermal solution, rather than from a previously existing mineral. We have added text on line 105 of the revised manuscript to clarify this.

109: There is hardly a consensus on this Johnson model of late evolution of oxygenic photosynthesis, and this mechanism can only explain surface oxidation of Mn, not within the top 1 km of crust.

Author response: We don't necessarily claim that the Johnson model must be correct, we merely propose it as a possible explanation for limited, early occurrences of oxidized Mn. We have added the word "possible" to line 131 of the revised manuscript to emphasize that we are simply suggesting a possible explanation that has appeared in recent literature.

120-121: Why are we not shown any plots for the this range of formation depths vs. age or Mn valence?

Author response: As noted in previous points, this is due to the fact that there is currently no depth data in the MED, and adding it for the many thousands of data points used in this study would have to be done by hand and would take many years of work. We are currently investigating the possibility of using machine-learning algorithms to extract this sort of data from the geologic literature in a more efficient fashion, and we look forward to being able to do these sorts of studies at some point in the future.

156: I will reiterate my point about a time lag requiring further process-based explanation with regards to minerals forming in equilibrium with their ambient redox conditions which are unlikely to reflect 66 Myr-old atmospheres at the time of mineralization.

Author response: We understand and appreciate the point the reviewer makes here. In point #5 above, we respond to this point and clarify what we mean, and we have added text on lines 187-193 of the revised manuscript to clarify this point for readers.

163: This subduction explanation is the only one for which mineralization could occur in an environment out of communication with the atmosphere for many Myr.

Author response: Indeed, we believe that subduction and re-melting is the primary way that oxidized material once near the surface mixes with deeper material, and slowly raises the overall redox state of the crust. We don't mean for the three mechanisms listed here to be mutually exclusive and competing explanations, we believe they are working in concert with each other to oxidize successively greater portions of the crust over geologic time. We have added the phrase "a combination of" to line 180 of the revised manuscript to make our meaning more clear to readers.

Reviewers' Comments:

Reviewer #1:

Remarks to the Author:

I think that the authors have addressed satisfactorily my comments in this revision. I recommend publication of this manuscript in Nature Communication with relatively minor revision.

- L. 143-144. The long-term increase is not visible after the GOE but only in the last 600 Ma. One can see a decrease of MnIII and Mn IV during Rodinia times that seems to sustain by a significant number of data.

- Concerning the authors' response: "As for why Mn oxidation state would be more sensitive to changes after 600 Ma than to changes during the GOE – we don't believe that it actually is. ... This preservation bias would therefore skew older data towards higher oxidation states, and dampen the apparent sensitivity of Mn oxidation state to atmospheric oxygen."

I don't see this mentioned in the main text. L. 133-136 concern the period before the GOE.

Reviewer #2:

Remarks to the Author:

I have read the revised manuscript, "The oxidation state of Earth's Crust: Evidence from the evolution of manganese minerals" by Daniel Hummer and co-authors. The authors have addressed some minor comments by myself and other reviewers and I commend them for these efforts, but they have declined to take action re: quantification and development of the proposed Mn mineral proxy to make the contribution novel and new. For this reason, I cannot recommend the revision for publication in Nature Communications.

The manuscript is well written and the topic is a hot one in Earth Sciences at this time. As it is currently written, the major contribution here is in assembling some specific finding from a large database. This is important, and the result is interesting, and I hope to see this published somewhere. The result of mining that database is confirmation that there are some variations in the average oxidation state of Mn contained in Earth's crust and this may be tied somehow to surface pO₂. This is not new, and it is my opinion that this work does not illuminate previously unknown or under-appreciated facets of our previous state of knowledge. I remain supportive of the authors' work on this topic, and in agreement with Reviewer #3, think that the new contribution these authors stand to make is in quantifying some of their broad claims. My collegial recommendation is that the authors invest time in developing this idea further and seek a longer format journal. If they wish to publish the story as finding from a database query, I recommend a short format article in a mineralogy journal.

Here is a summary of my specific comments on the revision, which I hope illuminate my concerns.

1. Abstract: "very few methods for evaluating crustal redox state". I disagree. All of the "numerous proxies" listed after line 45 that constrain fO₂ of the mantle and pO₂ of the atmosphere and oceans are done by quantifying the compositions of rocks that comprise and/or sample Earth's crust. Those proxies achieve what I, and reviewer #3 are recommending for these authors – quantitative links to a measure of oxygen in an environment which has led to the observed composition of the crust or crust-derived rocks.

2. The manuscript remains oversold. In the response to reviewers document, the authors rebut one of my comments in their point 1, stating that they have made a three-fold contribution to new knowledge by points a, b, and c. I agree that all points are important, but the claims that this manuscript is "first" is simply not true, and I disagree that the manuscript in its current form makes significant progress from the current state of knowledge. This is not meant to discourage the authors from their work. It is interesting and potentially important, but I think it can and must be developed beyond its current form of a database query to be published in a Nature journal.

3. In response to the authors rebuttal of my comments in their point 3, I do not doubt that my suggestion would take time to sort out properly. I remain convinced that this could be one path to addressing comments from myself and reviewer #3 re: quantification. In that same point, the authors clarify that they are speaking broadly of all processes that would lead to new Mn mineralization. I understand. I am suggesting that the authors work to pinpoint a more specific process that is important. Perhaps this effort would assist the authors in explaining why Archean-aged Mn minerals have the same average oxidation state as Phanerozoic Mn-minerals, despite a widely accepted ~5 order of magnitude increase in surface pO₂.

4. In response to the rebuttal of my comments re: citation, I point out to the authors that were they to seek a longer format journal, there would be adequate space for properly citing the work that precedes their own. Specifically, in response to their comment that it is clear that pO₂ is responsible for the proliferation of animal life in the Cambrian, and the range of other events that may also have played a role, I point the authors to an Annual Review in Earth and Planetary Science by Charles R Marshall in 2006. They will find that there is a rich body of literature on the topic that reflects the careful work of many people on the topic. For citations of how O₂ changed geochemical cycles, they may choose any of the papers they have already referenced in the context of how we know there have been fluctuations in pO₂ through time – disappearance of S-MIF, variations in U abundance of seafloor shales, etc. Scholarship is paramount, and should not be sacrificed to achieve a high-impact publication. In fact, I see a Nature Communications article in Earth Sciences, in the current issue online, with >50 citations.

I hope that this is helpful to the authors, and I wish them luck in making progress on this topic. If it remains in its current form, I recommend American Mineralogist.

Reviewer #3:

Remarks to the Author:

Hummer et al. have done excellent work in responding to my review of an earlier draft of their manuscript, and I am now comfortable in recommending that their article be accepted for publication in Nature Communications. With the new edits, this version of the manuscript has more clarity. I also appreciate the inclusion of several points as caveats and alternative interpretations at various points in their discussion. These qualifying points now make it clearer that the authors considered all the relevant scenarios and pieces of the puzzle for understanding their Mn mineral occurrence dataset as it stands at present.

I also thank the authors for their patience in providing clear, details responses to the points I raised in my original review. In particular, the information the authors provided on the work that has gone into assembling the database, and the reason why certain data is not currently available, helped convince me that the my requests for the displaying of quantitative relationships between formation depth and Mn mineral oxidation state were not reasonable requests at this point in time. This should not count against the authors as they have stressed that assembling depth data would require an unpractical workload for the associated payoff.

The authors asked that I provide detail on what further quantitative treatment of their dataset I felt was warranted, given that they did a good job of showing the statistical significance of their changes in Mn mineral oxidation state through time. The first of these points was the aforementioned plotting of data versus formation depth, which I have now been convinced was an unworkable request. The second area in which I felt quantitative detail was lacking in making estimates of average crustal fO₂ using this new Mn mineral proxy. With the addition to line 78 of an explicit reference to 'the redox state of Earth's crust (in terms of its average fO₂)' in the new manuscript, the reader is now led to expect a conclusion in terms of an average fO₂. I would not consider this an omission that should prevent the revised manuscript being published as-is, however it might strengthen the conclusions of the paper if even a simplified attempt to put a number on the variability in crustal fO₂ over time was made. I do not know whether this request is unreasonable and I will defer to the authors' expertise to decide whether such a calculation could be made and whether it could improve the impact of their eventual publication.

The above suggestion is just that, and I feel that the manuscript in its current form is now already suitable for publication in Nature Communications.

Below, we list reviewer comments individually and provide a response, also noting any changes we made to the manuscript in order to address the reviewer's concerns. All revisions to the manuscript are denoted with red font, and all responses to reviewer comments below are denoted in blue font. We are happy for these responses to be shown to reviewers should the editor deem it useful.

Reviewer 1:

1) L. 143-144. The long-term increase is not visible after the GOE but only in the last 600 Ma. One can see a decrease of MnIII and Mn IV during Rodinia times that seems to sustain by a significant number of data.

Author response: We thank the reviewer for pointing this out, we have specified the relevant time period in new text on line 147-148 of revision 2.

2) Concerning the authors' response: "As for why Mn oxidation state would be more sensitive to changes after 600 Ma than to changes during the GOE – we don't believe that it actually is. ... This preservation bias would therefore skew older data towards higher oxidation states, and dampen the apparent sensitivity of Mn oxidation state to atmospheric oxygen." I don't see this mentioned in the main text. L. 133-136 concern the period before the GOE.

Author response: Again, we thank the reviewer for attention to these details. We have added new text on lines 136-139 in revision 2 to clarify the connection between this type of preservation bias and the sensitivity of Mn redox state to oxygen in the post-GOE period.

Reviewer 2:

1) The result of mining that database is confirmation that there are some variations in the average oxidation state of Mn contained in Earth's crust and this may be tied somehow to surface pO₂. This is not new, and it is my opinion that this work does not illuminate previously unknown or under-appreciated facets of our previous state of knowledge.

Author response: We would like to reiterate that our Mn redox proxy and the conclusions it has enabled are completely new in the following respects: A) We have demonstrated that an entirely new type of proxy for redox conditions in the crust correlates with proxies for redox conditions in the atmosphere, which to the best of our knowledge has never been done before and the reviewer provides no evidence or examples besides the assertion that "This is not new". B) Our technique can be applied to other mineral systems to place new constraints on redox conditions in various Earth reservoirs, therefore making it applicable to answering multiple important questions in Earth system science, and C) We have documented for the first time the timing of oxidation of the crust vs. oxidation of the atmosphere – the details of this timing were previously unknown to the best of our knowledge. We contend that these are all important and novel contributions to our understanding of the coevolution of the geosphere and biosphere, and we don't see how the

reviewer's comments negates any of these points. If the reviewer is aware of previous publications that present a quantification of the averaged redox state of the entire shallow crust as a function of geologic time, we would be happy to cite these papers and include them in our discussion, but so far we cannot find any such studies and no one has pointed any out to us.

2) My collegial recommendation is that the authors invest time in developing this idea further and seek a longer format journal. If they wish to publish the story as finding from a database query, I recommend a short format article in a mineralogy journal.

Author response: We believe the reviewer is misunderstanding how this study was conducted. The “database query” that the reviewer refers to took the first author of this manuscript one full year of his life to conduct, format, and analyze because we are using extremely large quantities of data that are not always homogeneous in format or quality. The data must be carefully curated and assembled. There is no “database” one can simply “query” to conduct a study like this, which is why we present our data sources, criteria, and analytical methods in the manuscript. We understand that this is a very new approach to Earth systems science that not many researchers are acquainted with. But this is precisely why we claim we have developed a new redox proxy, and precisely why we wrote an article that describes it and submitted it to a prestigious scientific journal.

3) Abstract: “very few methods for evaluating crustal redox state”. I disagree. All of the “numerous proxies” listed after line 45 that constrain fO_2 of the mantle and pO_2 of the atmosphere and oceans are done by quantifying the compositions of rocks that comprise and/or sample Earth's crust. Those proxies achieve what I, and reviewer #3 are recommending for these authors – quantitative links to a measure of oxygen in an environment which has led to the observed composition of the crust or crust-derived rocks.

Author response: We understand the reviewer's point, and agree that this statement could potentially be misleading. To clarify what is novel about our study, we have revised the abstract to say “spatially averaged redox state of the crust”, to distinguish it from previous geochemical proxies that only apply to specific locations.

4) The manuscript remains oversold. In the response to reviewers document, the authors rebut one of my comments in their point 1, stating that they have made a three-fold contribution to new knowledge by points a, b, and c. I agree that all points are important, but the claims that this manuscript is “first” is simply not true, and I disagree that the manuscript in its current form makes significant progress from the current state of knowledge. This is not meant to discourage the authors from their work. It is interesting and potentially important, but I think it can and must be developed beyond its current form of a database query to be published in a Nature journal.

Author response: Frankly, we are puzzled by the reviewer's reaction. He/she states that our manuscript is not the first to do what we claim to do in points a, b, c, but he/she provides no examples and no specific rebuttal to a single one of these points. But science is not done by proclamation... in the scientific community, we examine evidence and cite sources when making claims about the state of our knowledge. Again, if the reviewer can cite examples of (A) a database of occurrences of redox sensitive minerals that are shown to correlate with proxies for atmospheric

O₂, or (B) a new redox proxy that can generate spatially averaged redox conditions and can be applied to many different mineral systems and reservoirs, or (C) a proxy that quantitatively establishes the relative timing between the oxidation of the shallow crust and oxidation of the atmosphere, we are more than happy to look at them and incorporate them into our manuscript. But we can find no such studies in the published literature, and we have no way of appropriately adjusting our claims or incorporating previous findings when the reviewer refuses to provide examples.

5) In response to the authors rebuttal of my comments in their point 3, I do not doubt that my suggestion would take time to sort out properly. I remain convinced that this could be one path to addressing comments from myself and reviewer #3 re: quantification. In that same point, the authors clarify that they are speaking broadly of all processes that would lead to new Mn mineralization. I understand. I am suggesting that the authors work to pinpoint a more specific process that is important. Perhaps this effort would assist the authors in explaining why Archean-aged Mn minerals have the same average oxidation state as Phanerozoic Mn-minerals, despite a widely accepted ~5 order of magnitude increase in surface pO₂.

Author response: We remain in agreement with the reviewer that an examination of the redox state of Mn minerals forming in different environments and through different paragenetic modes would bring great insight into the specific processes that oxidized the crust. However, as we explained previously, this would entail extensive and long term additions to the mineral evolution database and would necessitate a new study incorporating new data with completely different methods of analysis. In the meantime, we don't understand how this follow-up work, which is already underway (be on the lookout for a new paper by coauthor R.M. Hazen categorizing all mineral paragenetic modes), invalidates the global-scale study of Mn oxidation presented in the current manuscript. We certainly don't claim to have all the answers at this stage, what we claim is that a new type of redox proxy correlates with atmospheric pO₂ proxies, and reveals the timing between crustal and atmospheric oxidation on a global scale. Follow-up work on mechanisms can and should be done, but it would be helpful if the geoscience community is first shown how this new proxy works and what global-scale results it yields.

6) In response to the rebuttal of my comments re: citation, I point out to the authors that were they to seek a longer format journal, there would be adequate space for properly citing the work that precedes their own. Specifically, in response to their comment that it is clear that pO₂ is responsible for the proliferation of animal life in the Cambrian, and the range of other events that may also have played a role, I point the authors to an Annual Review in Earth and Planetary Science by Charles R Marshall in 2006. They will find that there is a rich body of literature on the topic that reflects the careful work of many people on the topic.

Author response: We are confused by this comment. Did something we say give the impression that we are unaware of work on changes in atmospheric pO₂ and its influence on animal life?? We are well aware that numerous researchers in the geological and biological sciences have contributed work on these issues for many decades, but our article is NOT a review article, it is an original research article that presents a new type of proxy for average crustal redox state. It would be neither feasible nor useful for us to summarize and cite the volumes of important work that has been done on these topics. Our goal in mentioning this connection was to establish, in the

introduction to our paper, that the redox evolution of Earth reservoirs is important to understand for many different reasons, one of which is its influence on biological evolution. We are certainly happy to cite Marshall (2006) as an example where this connection was discussed in the literature for the sake of ensuring that all our statements are properly supported. But expanding this one comment into an entire literature review on the Cambrian rise of O₂ and its influence on animal evolution is an unnecessary diversion from the focus of our manuscript.

7) For citations of how O₂ changed geochemical cycles, they may choose any of the papers they have already referenced in the context of how we know there have been fluctuations in pO₂ through time – disappearance of S-MIF, variations in U abundance of seafloor shales, etc. Scholarship is paramount, and should not be sacrificed to achieve a high-impact publication. In fact, I see a Nature Communications article in Earth Sciences, in the current issue online, with >50 citations. I hope that this is helpful to the authors, and I wish them luck in making progress on this topic. If it remains in its current form, I recommend American Mineralogist.

Author response: Every scientific article ever published provides an introduction of the topic, and explains its relevance to other subdisciplines and lines of research so that readers understand the context and motivation for the study. We do not see how the fact that the introduction section of our manuscript lacks a comprehensive review of all scientific literature could be reasonably interpreted as a lack of scholarship. Such reviews are the purview of book chapters and review articles. If the reviewer has comments about the quality of work presented in our manuscript, or is aware of related work that could potentially impact our findings, we are perfectly happy to hear them and engage in a meaningful dialogue. But quibbling that a simple introductory statement which explains the relevance of the work to a broad audience needs to be accompanied by a comprehensive literature review of all related subjects does not feel like a serious comment. Again, for the sake of being comprehensive in our citations, we have cited several articles from our citation list to support the statement that atmospheric O₂ changed geochemical cycles. Beyond this, we do not plan to delve further into those topics because we want the majority of our manuscript to focus on our research methods and results.

Reviewer 3:

First of all, we are very grateful to for this reviewer's many helpful insights and suggestions. We made many useful improvements to our manuscript based on these comments, and below we address the reviewer's one remaining suggestion.

1) With the addition to line 78 of an explicit reference to 'the redox state of Earth's crust (in terms of its average fO₂)' in the new manuscript, the reader is now led to expect a conclusion in terms of an average fO₂. I would not consider this an omission that should prevent the revised manuscript being published as-is, however it might strengthen the conclusions of the paper if even a simplified attempt to put a number on the variability in crustal fO₂ over time was made. I do not know whether this request is unreasonable and I will defer to the authors' expertise to decide whether such a calculation could be made and whether it could improve the impact of their eventual publication.

Author response: Computing an equivalent, average crustal fO_2 from our Mn redox data is an outstanding suggestion. Over the past many months, I have tried several different ways of doing such a computation, but each way relies on assuming a very particular geochemical system in which Mn minerals with different Mn oxidation states are in equilibrium with oxygen gas, such as the following:

One then computes the ratio of Mn^{4+}/Mn^{2+} from the redox data, and uses the equilibrium constant of the reaction to compute fO_2 . However, for this calculation to be valid one has to assume that these exact minerals are present in the system, AND that they are in direct equilibrium with molecular oxygen rather than an intermediary redox sensitive species.

Unfortunately, our dataset is composed of a wide variety of different Mn mineral species found in a wide variety of environments, and the thermodynamics of Mn oxides, sulfides, carbonates, silicates, and oxysalts are so different that no single set of assumptions can generate a self-consistent set of fO_2 values, because those values rely too heavily on the specifics of each geochemical system. We were therefore forced to conclude that conversion of the current dataset to fO_2 values is impossible, and that the average Mn redox state remains the best way to quantify our results.

However, we are genuinely grateful for this suggestion and want the reviewer to know that we made every attempt to make it work. But in the end, no single thermodynamic analysis could possibly be argued to be valid for such a wide array of mineral systems. With the addition of enough P, T, geochemical, and paragenetic data to the database, it may become possible to do these computations in the future. We are making progress on categorizing paragenetic modes and adding them to the database for individual localities, and we expect papers on this topic to be published in the next year.